# In Silico Analysis of Extended-Spectrum β-Lactamases in Bacteria

**DOI:** 10.3390/antibiotics10070812

**Published:** 2021-07-04

**Authors:** Patrik Mlynarcik, Hana Chudobova, Veronika Zdarska, Milan Kolar

**Affiliations:** 1Department of Microbiology, Faculty of Medicine and Dentistry, Palacky University Olomouc, Hnevotinska 3, 77515 Olomouc, Czech Republic; veronika.zdarska03@upol.cz (V.Z.); milan.kolar@fnol.cz (M.K.); 2Laboratory of Growth Regulators, Faculty of Science, Institute of Experimental Botany of the Czech Academy of Sciences, Palacky University, Šlechtitelů 27, 78371 Olomouc, Czech Republic; chudha01@upol.cz; 3Institute of Molecular and Translational Medicine, Faculty of Medicine and Dentistry, Palacky University Olomouc, Hnevotinska 5, 77900 Olomouc, Czech Republic

**Keywords:** ESBL, antibiotic resistance, bacteria, PCR, primer

## Abstract

The growing bacterial resistance to available β-lactam antibiotics is a very serious public health problem, especially due to the production of a wide range of β-lactamases. At present, clinically important bacteria are increasingly acquiring new elements of resistance to carbapenems and polymyxins, including extended-spectrum β-lactamases (ESBLs), carbapenemases and phosphoethanolamine transferases of the MCR type. These bacterial enzymes limit therapeutic options in human and veterinary medicine. It must be emphasized that there is a real risk of losing the ability to treat serious and life-threatening infections. The present study aimed to design specific oligonucleotides for rapid PCR detection of ESBL-encoding genes and in silico analysis of selected ESBL enzymes. A total of 58 primers were designed to detect 49 types of different ESBL genes. After comparing the amino acid sequences of ESBLs (CTX-M, SHV and TEM), phylogenetic trees were created based on the presence of conserved amino acids and homologous motifs. This study indicates that the proposed primers should be able to specifically detect more than 99.8% of all described ESBL enzymes. The results suggest that the in silico tested primers could be used for PCR to detect the presence of ESBL genes in various bacteria, as well as to monitor their spread.

## 1. Introduction

The mechanisms that bacterial pathogens have developed to fight antibiotics are many, one of them being production of enzymes degrading particular antibacterial agents. In case of β-lactam antibiotics, the enzymes are β-lactamases. To date, more than 7000 β-lactamases have been described in the Beta-Lactamase DataBase (BLDB) [1], which have different characteristics in terms of their substrate specificities. A special group is made up of β-lactamases with an extended spectrum of activity (extended-spectrum β-lactamases, ESBLs), which can inactivate broad-spectrum β-lactam antibiotics (e.g., penicillins, cephalosporins and monobactams). These enzymes are classified into Ambler molecular classes A and D; within the Bush-Jacoby-Medeiros classification system, they belong to groups 2be and 2d. While a common feature of class A ESBLs is their sensitivity to the activity of inhibitors such as clavulanic acid, sulbactam and tazobactam, class D β-lactamases (oxacillinases; OXA) are resistant to them. Furthermore, class C ESBLs have also been described (e.g., ADC). Some may also hydrolyze carbapenems, like the chromosomally encoded ADC-68 enzyme described in *Acinetobacter baumannii* (*A. baumannii*), whose R2 and C-loops allow better accommodation of carbapenems [2]. Moreover, weak hydrolytic activity against carbapenems was also described for some other variants such as ACT-1, ACT-28, CMY-2 or CMY-10 (AmpC type β-lactamase) [3,4].

Regarding OXA enzymes, an OXA-23 subfamily variant (OXA-146 with an alanine duplication at position 220) possessing both ESBL and carbapenem-hydrolyzing class D β-lactamase properties has been described [5]. Recently, two novel ESBL genes probably associated with small mobilizable plasmids, named *bla*_RSA1_ and *bla*_RSA2_, have been found in Indian river sediments, the latter also showing weak carbapenemase activity [6]. In addition, increased carbapenem resistance may be observed due to the production of ESBL enzymes if some secondary resistant mechanisms such as porin loss are present [7].

A great proportion of ESBLs are TEM (Temoneira class A extended-spectrum β-lactamase) or SHV (sulfhydryl variant of the TEM enzyme) enzyme derivatives. However, the most widespread enzymes are CTX-M (cefotaxime-hydrolyzing β-lactamase–Munich) whose production and dissemination rates have increased significantly since the mid-1990s [8]. Newly discovered and, in terms of occurrence, unusual class A ESBL enzymes include BEL, BES, GES (Guiana extended-spectrum β-lactamase), PER, VEB, SFO and TLA β-lactamases. In addition, the ESBL phenotype has been reported in other groups of β-lactamases such as CARB and L2. Additional minor ESBL types, such as FONA, BPU and YOC, have also been recently identified [9,10,11]. An interesting example of the minor ESBL group is SFO-1 encoded by a plasmid which includes the *ampR* regulatory gene, which allows for β-lactamase induction in a manner similar to class C β-lactamases. Minor ESBL enzymes are rare but often can associate with resistance genes against other antibiotics such as aminoglycosides or quinolones [12,13,14].

Many ESBL genes have been identified as a source of acquired resistance, but further studies show that ESBLs also occur naturally in clinically relevant pathogens and in environmental species. For example, many chromosomally encoded and naturally occurring ESBLs, such as CSP-1, KLUA-1, KLUC-1, OXY-1 RAHN-1 and SGM-1, have been described in various bacteria, although their role in phenotypic resistance is small [15,16]. Other enzymes with the ESBL phenotype include chromosomally encoded metallo-β-lactamases (MBLs), HMB-1, as well as KHM-1 found in multidrug-resistant bacteria [17].

GES enzymes were first classified as ESBLs due to a large number of hydrolyzable substrates including penicillins and cephalosporins with an extended spectrum. Since GES enzymes hydrolyzed, to a lesser extent, imipenem as well, they were also included among group 2f carbapenemases. Genes encoding this enzyme family can be located in integrons on plasmids. Even though they are rare, there have been reports about their presence worldwide [18].

Most SHV enzymes with an extended spectrum are derived from SHV-1 with differences in one or more amino acids; this small change is sufficient to create an extended spectrum. In particular, it is the G238S or E240K substitutions, with a serine residue being responsible for ceftazidime hydrolysis and a lysine residue for cefotaxime cleavage (both third-generation cephalosporins) [12]. Carbapenems do not belong to the hydrolytic profile of SHV enzymes, but there was a clinical isolate of *Klebsiella pneumoniae* (*K. pneumoniae*) with low sensitivity to some extended-spectrum cephalosporins as well as to imipenem. Subsequently, it has been found that a β-lactamase variant of SHV-1 with amino acid change A146V, designated SHV-38, is responsible for reduced imipenem sensitivity. Genes encoding SHV-38 are located on chromosomes, which means that it is the first SHV chromosomal enzyme with an extended spectrum of hydrolysis [19]. CTX-M enzymes do not have the ability to hydrolyze carbapenems. However, relatively recently identified CTX-M-33, a derivative of the globally widespread enzyme CTX-M-15 differing only in one amino acid substitution (N109S), exhibited the ability to hydrolyze meropenem. This property results from the substitution of N109S and strong selection of the antibiotic [20].

Enterobacteriaceae members are the main ESBL producers and they have mainly been recorded in hospital and community environments. However, the presence of ESBL producers (mainly CTX-M-15) has also been shown in foodstuffs of animal origin (e.g., cow’s milk, dairy products, chicken), suggesting possible transfer through the food chain [21,22]. In addition, an increasing number of ESBL-producing enterobacteria isolated from water environments have been observed [23]. In this case, a total of 10 ESBL-positive isolates [nine *Escherichia coli* (*E. coli*) and one *K. pneumoniae*] were identified in four well waters out of 100. ESBL genotyping revealed that CTX-M-15 was present nine times and CTX-M-27 was produced once. This and many other studies suggest that, for example, ESBL-producing enterobacteria in rural waters can spread to animals and humans via drinking water.

In view of the growing clinical importance of ESBL enzymes, their detection is also necessary in routine microbiological practice. The present study was concerned with (1) in silico analysis of ESBL enzymes, and (2) designing primers serving to detect all described ESBL genes.

## 2. Results

### 2.1. In Silico Analysis of ESBL Enzymes

A search of the BLDB and BLASTn databases showed that a wide variety of class A and C β-lactamases, MBLs and class D β-lactamases such as OXA have been described for both enterobacteria and Gram-negative non-fermenting bacteria. The study of selected clinically significant ESBL enzymes (CTX-M, SHV, TEM) revealed that the most widespread group in bacterial genera were ESBL enzymes of the TEM type found in 45 genera, followed by CTX-M in 25 and SHV in 23 genera (Table 1). In addition, ESBL enzymes of the CTX-M, GES, OXA (OXA-1-like, OXA-2-like, OXA-10-like), SHV, TEM and VEB types are most commonly found in selected enterobacteria (*Enterobacter* spp., *Escherichia* spp., *Klebsiella* spp.) and Gram-negative non-fermenting rods (*Acinetobacter* spp., *Pseudomonas* spp.; data not shown).

In silico analysis of ESBL enzymes was used to detect the presence of various conserved amino acids and motifs. Several conserved amino acids such as methionine-cysteine-serine-threonine-serine-lysine at positions 71–76 (71-MCSTSK-76; numbering according to CTX-M-1; Figure 1A) were identified within studied CTX-M enzymes. In this figure, there are no remaining amino acid sections and amino acids such as methionine at position 1 (M1), glutamine at positions 34, 35 and 268 (Q34, Q35 and Q268), leucine at position 37 (L37), alanine and glutamic acid at positions 273–274 (273-AE-274), leucine and alanine at positions 280–281 (280-LA-281) and alanine at position 284 (A284).

Comparison of the amino acid sequences of the selected CTX-M showed the same (55.3–99.7%) sequence identity between them. For example, CTX-M-150 and CTX-M-151 had only 55.3% amino acid identity, whereas for CTX-M-155 and CTX-M-157, it was 99.7% identity (results not shown).

Reconstruction of the phylogenetic tree allowed monitoring of the affinity of individual amino acid sequences of CTX-M enzymes. Overall, 216 types of different CTX-M were included in the analysis. CTX-M enzymes were divided into several main groups and subgroups, see Figure 2).

By comparing the 199 sequences of SHV enzymes, individually conserved motifs of active sites including X-X-phenylalanine-lysine were identified at positions 66–69 (66-XXFK-69; numbering according to SHV-1; Figure 1B). Other amino acid sections and amino acids which are not shown in the figure include arginine at positions 2 and 5 (R2, R5), isoleucine at positions 8 to 9 (I8, I9), leucine at positions 11 to 12, 17 and 26 (11-LL-12, L17, L26), valine at position 19 (V19), serine-proline-glutamine at positions 22–24 (22-SPQ-24), glutamine-isoleucine-lysine at positions 28–30 (28-QIK-30), serine and glutamic acid at positions 32–33 (32-SE-33), serine and glycine at positions 37–38 (37-SG-38), methionine at position 266 (M266), glutamine at position 271 (Q271), isoleucine and alanine at positions 273–274 (273-IA-274), glycine at position 277 (G277), alanine at position 279 (A279) and glutamic acid-histidine-tryptophan-glutamine at positions 282–285 (282-EHWQ-285).

In this case, the comparison of the amino acid sequences of SHV enzymes showed 93.1–99.7% sequence identity between them. For example, SHV-16 and SHV-100 had 93.1% amino acid consensus, while in the case of SHV-7 and SHV-105, the identity was 99.7% (results not shown).

The relationships of individual SHV enzymes were shown using a phylogenetic tree. A rooted phylogenetic tree enabled us to distinguish different clusters and identify several major groups and subgroups (Figure 3).

Conserved amino acids and motifs including serine-threonine-phenylalanine-lysine at positions 68–71 (68-STFK-71; numbering according to TEM-1; Figure 1C) were identified by analysis of another 199 sequences of TEM enzymes. The remaining identical amino acid sections and amino acids which are not shown in the figure include methionine at position 1 (M1), histidine-phenylalanine-arginine-valine at positions 5–8 (5-HFRV-8), proline and phenylalanine at positions 12–13 (12-PF-13), alanine-alanine-phenylalanine-cysteine at positions 15–18 (15-AAFC-18), proline-valine-phenylalanine at positions 20–22 (20-PVF-22), proline at position 25 (P25), threonine at position 27 (T27), valine-lysine-valine at positions 29–31 (29-VKV-31), alanine and glutamic acid at positions 34–35 (34-AE-35), glutamic acid-isoleucine-glycine at positions 277–279 (277-EIG-279), and serine-leucine-isoleucine-lysine at positions 281–284 (281-SLIK-284).

The amino acid consensus of the studied sequences of TEM enzymes were in the range of 93.4–99.7%. The lowest 93.4% amino acid consensus was recorded, for example, at TEM-178 and TEM-194, with 99.7% consensus, for example, between TEM-189 and TEM-191 (results are not shown).

By using a phylogenetic tree, relationships among TEM enzymes were shown based on the similarity of amino acid sequences. A rooted phylogenetic tree allowed us to observe different clusters and identify several major groups and subgroups (Figure 4).

Point mutation studies showed that the highest numbers of amino acid changes were observed in the CTX-M, OXY, TEM, SHV, PER and VEB types, namely 6344, 904, 534, 418, 162 and 61, respectively (Table 2). In case of CTX-M enzymes, the most common amino acid changes recorded were as follows: S → T (275 times, this amino acid change was also described in all remaining ESBL enzymes), K → Q (217 times) and Q → R (198 times). Conversely, amino acid changes recorded only once were A → I, M, Q or R, etc. The most common amino acid changes in the remaining ESBL enzymes included A → T (108 times), I → V (15 times), L → Q (73 times), E → K (82 times) and I → V (14 times) in OXY, PER, SHV, TEM and VEB, respectively.

### 2.2. Detection of ESBL-Positive Bacteria with PCR

A total of 49 different types of ESBLs were analyzed in this study, including 1438 β-lactamase genes, of which only 624 enzymes had ESBL phenotype based on BLDB database and literature search. There were 42 members of class A β-lactamases (e.g., CARB, CTX-M, GES, OXY, PER, SHV, TEM, VEB), two members of the class C β-lactamase family (ADC and YOC), two gene families of the subclass B1 (HMB and KHM), and three members of class D β-lactamases (BPU, CDD and OXA; 5 OXA subgroups, OXA-18 and OXA-45; Table 3). In case of the most numerous groups of ESBL enzymes (CTX-M, SHV, TEM), there were 216, 199 and 199 variants of these enzymes with GenBank accession numbers, respectively. The CTX-M enzymes were subclassified based on similarity of amino acid sequences into 5 groups (CTX-M-1-like, CTX-M-2-like, CTX-M-8-like, CTX-M-9-like and CTX-M-151-like). A total of 58 specific primer pairs for ESBL detection were designed (Table 3) in silico using Primer3 (Geneious, Biomatters). The primer-BLAST results showed that these primers could detect all or almost all allelic variants of these so far described types of enzymes that are commonly found in bacteria.

## 3. Discussion

Currently, clinically significant bacteria are increasingly resistant to carbapenems and polymyxins due to the production of ESBL, carbapenemases and phosphoethanolamine transferases of the MCR type, which can overcome the last-line antibiotics for the treatment of infections caused by multidrug-resistant Gram-negative bacterial pathogens [24,25,26].

For the past decade, we have observed a global rapid increase in ESBL- and carbapenemase-producing bacteria in animals intended for food production [27]. Other sources of transmission and dissemination of β-lactamases include, besides hospital or community environments, also coins and paper currency. An example is the presence of various Gram-negative bacteria producing CTX-M-type ESBLs and OXA-48 carbapenemase on the surface of Algerian currency. Especially simultaneous manipulation of money and food contributes to the spread of infectious agents and thus of bacterial resistance to antibiotics [28].

ESBLs have become a global problem in the treatment of hospitalized patients after the introduction of β-lactam antibiotics with an extended spectrum of activity into clinical practice. Most microorganisms producing these enzymes belong to the Enterobacteriaceae family, the most widespread producers being *K. pneumoniae* and *E. coli* isolated from the hospital environment. ESBL-producing bacterial strains are most commonly found in hospital patients, with the risk factors being prolonged hospital stay, disease severity, previous exposure to antibiotics, time spent in intensive care or presence of a urinary/arterial catheter [12,29].

The ever-expanding problem of ESBL resistance is largely due to frequent and unjustified prescription, especially of broad-spectrum cephalosporins. Increasing resistance to third-generation cephalosporins in *E. coli* isolates can be observed in a gradient from northern to southern countries, with the lowest and highest percentage in Northern and Southern Europe, respectively. The current resistance status can be evaluated using the EARS-NET database. For example, the percentage of *E. coli* isolates resistant to third-generation cephalosporins in 2019 was 7.8% in Sweden, compared to 30.9% in Italy. In case of *K. pneumoniae*, resistance to third-generation cephalosporins in most European countries was below 60%, but exceeded 70% in some countries such as Bulgaria (data obtained from the EARS-NET database; https://ecdc.europa.eu/en/antimicrobial-resistance/surveillance-and-disease-data/data-ecdc, accessed on 31 March 2021).

By the end of the 1990s, the majority of identified SHV- and TEM-type ESBLs originated from nosocomial isolates of *K. pneumoniae* (especially in intensive care units). In the following years, the epidemiological situation concerning ESBLs underwent dramatic changes. The main producers of ESBLs were *E. coli* strains expressing CTX-M-type β-lactamases that spread mainly through mobile genetic elements. There was also an increase in the number of isolates from the community, mostly from patients with urinary infection [30].

One of the objectives of the present study was comparison of amino acid sequences of the investigated ESBL enzymes to elucidate the conserved amino acids and motifs (sections). The presence of various conserved motifs is discussed by many authors [31]. The present study showed a large number of conserved motifs and amino acid residues in the selected ESBL enzymes (Figure 1A–C). The seemingly smaller amount of conserved amino acid residues and sections may be primarily related to the use of all variants of the studied ESBL types. Within the analyzed types of ESBL enzymes, the lowest amino acid identity was found for CTX-M enzymes, ranging from 55.3 to 99.7%, which also correlates with the highest number of point mutations (Table 2) found in this enzyme group.

Sometimes it is not easy to classify some β-lactamases into subgroups based on sequential and other properties because they are rather varied. Phylogenetic trees were created (Figure 2, Figure 3 and Figure 4), which show similarity of the studied subtypes of ESBL enzymes. Obviously, there is a great variety between genes encoding individual types of ESBL.

In general, many proposed primers or methods described in the literature are generally useful for detecting the most common types of ESBL but do not cover all variants of these genes. Therefore, the present study aimed to design specific primers for rapid PCR detection of all known ESBL genes described in the BLDB database. Briefly, our results suggest that proposed primers (58 primer pairs; Table 3) have the ability to specifically detect 623 analyzed ESBL subtypes (99.8%) and are suitable for detection and epidemiological analysis of all described ESBL genes in various bacteria. Additional primers for detection of β-lactamase genes, including carbapenemases [e.g., *bla*_KPC_ (*Klebsiella pneumoniae* carbapenemase), *bla*_NDM_ (New Delhi MBL), *bla*_VIM_ (Verona integron-encoded MBL)] and OXA subgroups, are described elsewhere [27,32,33,34].

The CARB-F2/R2 primers were designed to detect seven CARB subtypes, including CARB-10 (ESBL) enzyme. A total of 51 subtypes of these enzymes have been described. The remaining subtypes of CARB enzymes except for CARB-42 could be verified using the specific primers described previously [34]. This could be detected with an additional primer (F-TCTCTCCTCGAGCAACAAA, R-AAGTGAGAGCTCGGTTTCT; Tm: 53, PCR product: 708 bp). The L2-F1/R1 primers can be used to detect 18 L2 subtypes, including four L2-1/-2/-3/-E-10 (ESBL) enzymes, while not being able to distinguish 4 subtypes. The remaining L2 subtypes can be detected using the L2-F2/R2 primers [34]. To confirm the presence of the *bla*_KLUB_, *bla*_KLUG_ and *bla*_KLUY_ genes, the primers CTX-M-F1/R1 could be used to detect all two, five and five different subtypes listed in the BLDB database, respectively. The primers CTX-M-F/R were used to detect 215 subtypes of *bla*_CTX-M-like_ genes but were unable to distinguish one allelic variant (CTX-M-151). Thus, a primer (F-CAGTAAAGTGATGGCGGTAG, R–ATACCACGGCAATATCGTTG; Tm: 54 °C, PCR product: 536 bp) could be used to detect it. The primers ADC-F/R (ADC-F1/R1 to ADC-F4/R4) were used to detect 31 ESBL subtypes (e.g., ADC-33, ADC-117, ADC-140) of the *bla*_ADC-like_ genes. The remaining 12 subtypes of ADC enzymes could be detected using the specific primers described elsewhere [34]. The OXA(7)-F/R primer was aimed at detecting 38 subtypes of the *bla*_OXA-48-like_ genes, including ESBL types OXA-163 and OXA-405. To date, a total of 41 subtypes of OXA-48-like enzymes have been described in the BLDB database. Another 3 specific OXA-48-like subtypes (OXA-436/-535/-731) could be tested using the primer (F-ACGAGAATAMACAGCAGGG, R-GATAMACAGGCACAACCGA; Tm: 56, PCR product: 228 bp).

We are currently witnessing increasing antibiotic resistance in clinically important bacteria, which is associated with the discovery of new enzymes that break down antibiotics, in our case ESBLs, which appear over time. In some cases, very numerous variants of these enzymes appear, which can significantly limit their accurate detection. Therefore, continuous analysis of all known ESBL enzymes and design of more specific primers is necessary to prevent their spread. For example, OXA enzymes represent a rapidly growing family that includes over 943 enzymes [1] that are highly diverse in terms of sequence. However, in case of OXA enzymes, another difficulty is their accurate and timely detection, since OXA-encoding genes are expressed only in the presence of functional promoters represented by insertion sequences. Another very large group is, for example, AmpC β-lactamases *bla*_EC_ (formerly *bla*_ESC_, chromosomally encoded cephalosporinases), in which more than 2200 variants have been described and their number is growing. Although these are only chromosomally encoded β-lactamases, some exhibit the ESBL phenotype and, together with overexpression of efflux pumps and low outer membrane permeability, they are increasingly reported with regard to multidrug resistance, for example in *A. baumannii* [35]. Ultimately, this suggests that antibiotic resistance in bacteria is a complex phenomenon.

Authors often state that broad-spectrum cephalosporin-resistant isolates are ESBL-negative [36]. Therefore, we cannot rule out the possibility that these bacterial strains also contain other rare ESBL types such as CARB, KLUC or very rare OXA variants. Furthermore, we must also consider other resistance mechanisms, such as over-production of chromosomal AmpC, increased expression of an efflux pump or reduced permeability of the outer membrane, with new and non-described resistance mechanisms not being excluded. The solution seems to be transcriptome sequencing, as well as other forms of sequencing, such as whole genome sequencing, which will provide new possibilities for resistance prediction in the near future.

## 4. Materials and Methods

### 4.1. Sequence Analysis

A total of 624 sequences of genes encoding ESBLs (with definitive assignment) described in the BLDB (http://bldb.eu; last accessed in 31 May 2021) [1] were downloaded from the GenBank database. Comparison of nucleotide/amino acid sequences and mutation analysis were performed using the bioinformatics software Geneious Prime (Biomatters). Multiple sequence alignments were carried out using the default settings of the Geneious alignment algorithm (cost matrix: 51% similarity; gap open penalty: 12; and gap extension penalty: 3) to identify highly homologous regions suitable for designing primers.

### 4.2. Phylogenetic Tree Construction

The phylogenetic tree was made by Geneious software using PhyML based on the Le and Gascuel model. The first phylogenetic tree was obtained by comparing 216 various types of CTX-M enzymes; the second and third ones consist of 199 SHV and 199 TEM β-lactamases, respectively.

### 4.3. Designing Primers for PCR

Homologous regions in nucleotide sequences were used for designing primers with Primer3 (Geneious Prime) with the following requirements: an optimal melting temperature of 52–60 °C, a GC content varying from 40% to 60%, an optimal oligo length between 17 and 22 base pairs, and an amplification product size of 225 to 820 base pairs. All the oligonucleotides were tested in silico for hybridization with ESBL genes contained in the BLDB database. The primer specifications are listed in Table 3.

## 5. Conclusions

The present study reports 58 in silico and in vitro tested primer pairs for PCR assay that may be able to distinguish 99.8% of ESBL-producing bacteria. These may be part of diagnostic tests for the detection of observed resistance genes in bacterial pathogens. Such diagnostic tests can be used for early detection, monitoring and dissemination of ESBLs, thus contributing to reducing the spread of ESBL-positive bacteria, adequate antibiotic treatment and reducing health care costs.

## Figures and Tables

**Figure 1 antibiotics-10-00812-f001:**
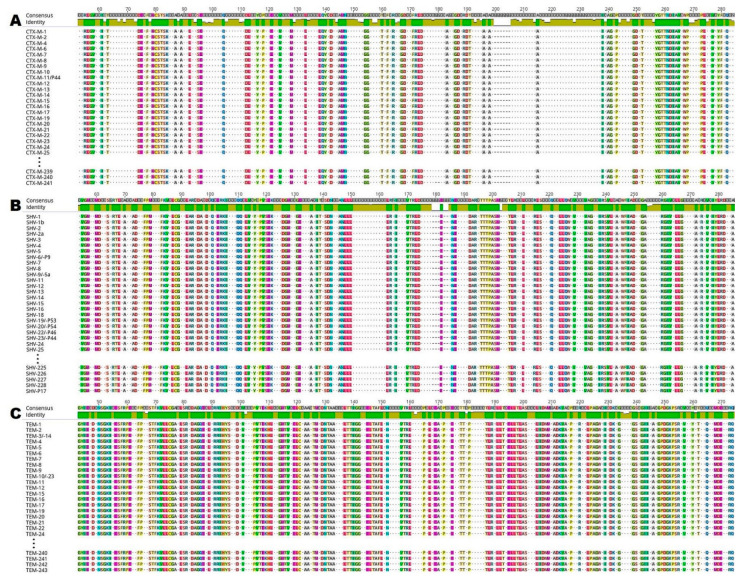
Comparison of 216, 199 and 199 amino acid sequences of (**A**) CTX-M (cefotaxime-hydrolyzing β-lactamase–Munich); (**B**) SHV (sulfhydryl variant of the TEM enzyme); and (**C**) TEM (Temoneira class A extended-spectrum β-lactamase enzymes), respectively. The analysis and image creation were performed in Geneious Prime. Only the same amino acid residues in all sequences are highlighted in the figure. The green panel indicates amino acids that are identical in all sequences on a given position. Different amino acids are marked with dots.

**Figure 2 antibiotics-10-00812-f002:**
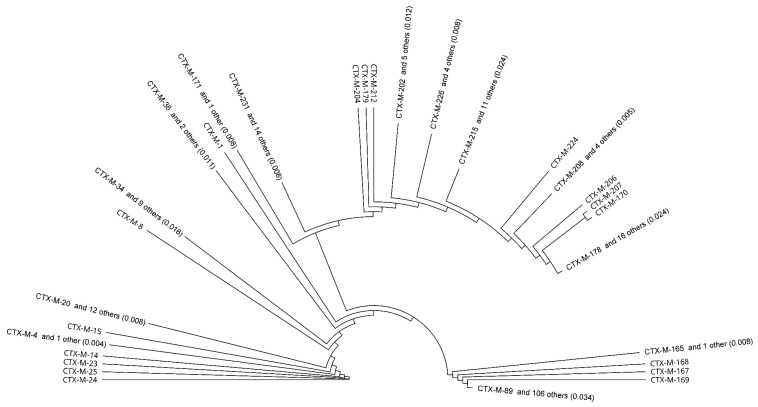
A phylogenetic tree obtained by comparing 216 CTX-M enzymes using Geneious PhyM and automatic subtree compression (subtree distance 0.039).

**Figure 3 antibiotics-10-00812-f003:**
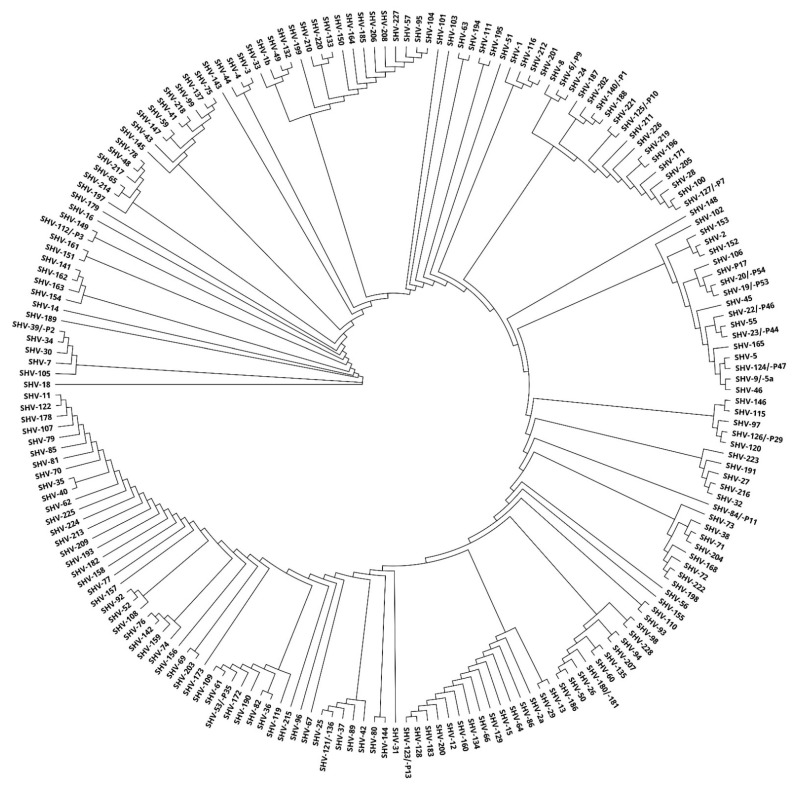
A phylogenetic tree obtained by comparing 199 SHV enzymes using Geneious PhyML.

**Figure 4 antibiotics-10-00812-f004:**
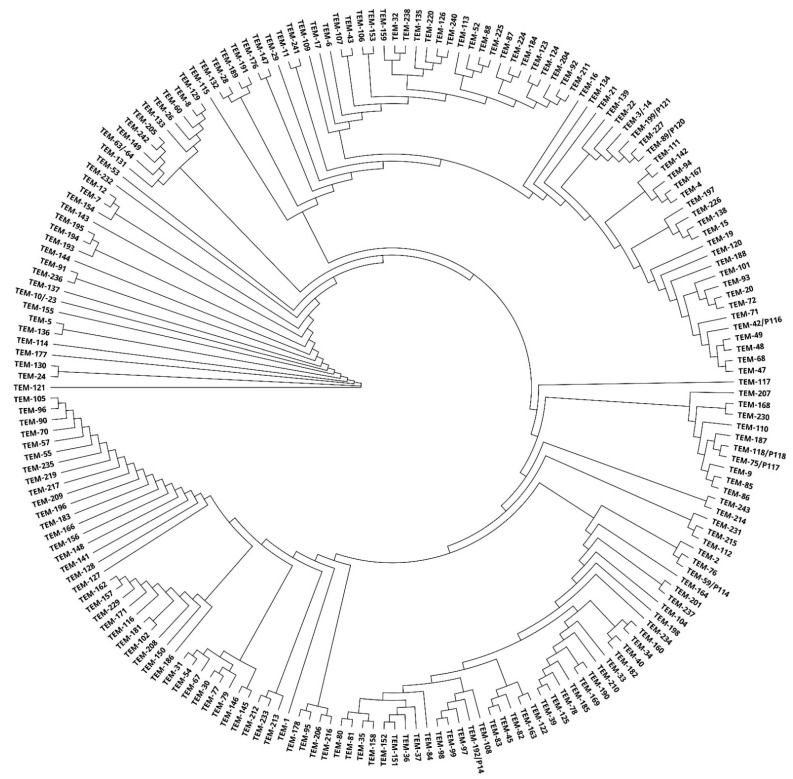
A phylogenetic tree obtained by comparing 199 TEM enzymes using Geneious PhyML.

**Table 1 antibiotics-10-00812-t001:** Distribution of (extended-spectrum β-lactamases) ESBL enzymes in bacterial genera.

Bacterial Genera	ESBL Enzymes	Bacterial Genera	ESBL Enzymes
*Achromobacter*	TEM	*Legionella*	TEM
*Acinetobacter*	CTX-M, SHV, TEM	*Lysobacter*	TEM
*Aeromonas*	CTX-M, SHV, TEM	*Morganella*	CTX-M, SHV, TEM
*Alcaligenes*	TEM	*Mycobacterium*	TEM
*Atlantibacter*	CTX-M, TEM	*Neisseia*	TEM
*Bacillus*	CTX-M, SHV, TEM	*Nocardia*	TEM
*Bordetella*	TEM	*Ochrobactrum*	CTX-M, SHV
*Brevibacterium*	CTX-M, TEM	*Pantoea*	CTX-M, TEM
*Brucella*	TEM	*Pasteurella*	TEM
*Burkholderia*	CTX-M, SHV	*Proteus*	CTX-M, SHV, TEM
*Campylobacter*	TEM	*Providencia*	CTX-M, TEM
*Cedecea*	SHV	*Pseudomonas*	CTX-M, SHV, TEM
*Citrobacter*	CTX-M, SHV, TEM	*Pseudochrobactrum*	CTX-M
*Cronobacter*	SHV, TEM	*Ralstonia*	TEM
*Elizabethkingia (formerly Chryseobacterium)*	TEM	*Raoultella*	CTX-M, SHV, TEM
*Enterobacter*	CTX-M, SHV, TEM	*Salmonella*	CTX-M, SHV, TEM
*Enterococcus*	SHV, TEM	*Serratia*	CTX-M, SHV, TEM
*Erwinia*	TEM	*Shewanella*	CTX-M
*Escherichia*	CTX-M, SHV, TEM	*Shigella*	CTX-M, SHV, TEM
*Haemophilus*	TEM	*Staphylococcus*	TEM
*Hafnia*	TEM	*Stenotrophomonas*	SHV
*Kerstersia*	SHV	*Streptococcus*	TEM
*Klebsiella*	CTX-M, SHV, TEM	*Streptomyces*	TEM
*Kluyvera*	CTX-M, SHV, TEM	*Superficieibacter*	SHV, TEM
*Lactobacillus*	TEM	*Vibrio*	CTX-M, TEM
*Leclercia*	CTX-M, TEM	*Yersinia*	TEM

**Table 2 antibiotics-10-00812-t002:** Point mutations in selected ESBL enzymes.

Amino Acid Change		Number of Mutations (Amino Acid Position)					
from	to	CTX-M *	OXY *	PER *	SHV *	TEM *	VEB *
A	C		7 (22)	2 (7)			
A	D	36 (32/131/157)		3 (114)	1 (264)	1 (230)	
A	E	28 (35/46/157/222/236)	21 (230)	2 (114/166)			1 (38)
A	G	158 (16/64/73/131/136/157/297)	10 (240)	1 (7)	3 (205/259)	2 (235/280)	
A	I	1 (16)					
A	K	22 (64/244)					
A	M	1 (73)					
A	N	36 (35)					
A	P	78 (59/78/272)			1 (15)	1 (183)	
A	Q	1 (157)		2 (151)			
A	R	1 (222)			1 (151)	1 (185)	
A	S	102 (35/46/151/235/244)	2 (240)	4 (109)	1 (23)	3 (9/40/266)	
A	T	110 (16/18/35/90/120/131/137/161/222/244)	108 (16/30/32/38/56/129/171/208/240/284)	1 (11)	14 (20/23/63/90/125/135/157/161/204)	12 (9/183/235/280)	4 (248)
A	V	74 (16/35/88/131/135/137/163)	32 (16/233)		20 (20/63/125/135/137/145/157/188/205/218/264/288)	18 (23/40/182/222/266/276)	
A	W				1 (299)		
C	N				1 (9)		
C	R				1 (134)		
C	S				1 (9)		
D	A	69 (36)	1 (39)	4 (155)	3 (195/231)		
D	E	160 (218/303)	1 (242)		2 (168/230)	2 (155/177)	
D	G	179 (259/270)			4 (115/195/230/231)	2 (113/161)	
D	H	23 (36/292/303)			2 (112)	1 (161)	
D	K	2 (303)					
D	N	40 (36/218/270/303)	28 (39/58)		1 (195)	3 (36/113/174)	
D	P	1 (292)				1 (33)	
D	Q	1 (218)					
D	S	25 (36/65/256/303)					
E	A	106 (47/177)					
E	D	35 (98/169)	22 (92/99/149/278)		2 (29/273)	1 (26)	
E	G	1 (47)		1 (33)	3 (59/75/177)	2 (164/237)	
E	K	3 (47/132/169)	1 (278)		37 (29/100/256/289)	82 (26/62/102/166/237)	
E	N	1 (169)					
E	P	1 (107)					
E	Q	85 (98/132)	4 (92)	8 (119/193/224)	1 (75)		
E	R				1 (256)	1 (237)	
E	T	1 (98)					
E	V			3 (23)		1 (237)	
F	A	10 (12)					
F	L	1 (12)		3 (19)		3 (14/228)	
F	M	72 (12)					
F	S	21 (12)			1 (162)		
F	W			1 (22)			
F	Y						2 (64)
G	A	31 (25/217)		4 (149/190)	6 (158/191/255)		
G	C	3 (47/132/169)			1 (258)		
G	D		1 (159)		11 (98/154/167/245)	4 (39/90/194/236)	
G	E	1 (25)		1 (41)	2 (158/296)	1 (216)	
G	H	1 (304)					
G	K	1 (126)					
G	N	1 (53)				1 (236)	
G	R	4 (54/234/304)			1 (296)	2 (154/236)	
G	S	27 (25/213/234/253/304)	30 (24/75)		42 (65/155/167/255)	40 (194/236)	
G	V					2 (194/263)	
H	D			2 (134/154)			
H	F	1 (123)					
H	K	1 (152)					
H	L			2 (134)		4 (285)	
H	N					1 (156)	
H	Q	74 (152)			1 (22)		
H	R					5 (24/151)	
H	T				1 (107)		
H	Y				3 (123)		
I	A	1 (69)		2 (5/97)			
I	F	1 (119)			11 (6)	1 (11)	
I	L	1 (150)	5 (33)	3 (61/194)	2 (297/302)		
I	M			4 (111)	1 (275)	1 (259)	
I	P					4 (3)	
I	R					1 (11)	
I	S	1 (300)					
I	T	13 (69)		7 (5/40/97)	2 (275/297)	1 (275)	1 (18)
I	V	170 (69/108/266/275)	7 (111)	15 (61/162/194/245/258)	3 (6/58)	5 (125/171/256)	14 (18)
K	A	5 (109/110/299)	23 (102)	3 (102)			
K	D	1 (214)					
K	E	36 (93/122/269)	27 (180)	1 (113)		1 (32)	
K	G	1 (214)					
K	H	73 (4/214)					
K	L	1 (148)	20 (140)				
K	N	80 (3/214/286)	20 (101)		2 (209/271)		
K	P	81 (110)					
K	Q	217 (4/44/94/109/269/286)		2 (196)		1 (144)	
K	R	103 (3/44/94/109/122/229/251/299)		7 (196/246)	5 (251/271)		2 (237)
K	S	7 (110/214)					
K	T	91 (3/269)				1 (190)	
L	A	4 (9/14/23/210)		2 (256)			
L	F	121 (14/24/30/130/305)	1 (84)		6 (124/133/189)	23 (19/55/100/136)	1 (56)
L	G	2 (30/70)					
L	I	111 (9/149/166/276)	24 (2)	1 (256)		4 (19/135/218)	
L	M	5 (9/22/102/207/242)	23 (2/11)	4 (75/260)	2 (17/237)	3 (47/219)	
L	N	1 (305)					
L	P	4 (130/149/212)	1 (262)		4 (38/62/102/301)	1 (49)	
L	Q	2 (60/180)			73 (33/62)		
L	R				5 (8/102/133/185)	1 (10)	
L	S	3 (14/23/305)					
L	V	193 (9/60/101/153/216)		3 (305)	2 (159/210)	4 (28/38/100/247)	
L	Y	15 (224/305)					
M	A	69 (14)					
M	G	1 (197)					
M	I	1 (86)	35 (12/141)	1 (160)	3 (80)	10 (66/67/153/180)	
M	K				4 (3)		
M	L	105 (228)	20 (12)	3 (160)	1 (80)	17 (67)	
M	T	21 (14)				34 (180)	
M	V				7 (80/140/228)	11 (67/184)	
N	D	89 (62/125/143)	36 (54/90/200/256)	3 (308)	2 (269/291)	12 (272)	6 (294)
N	E	1 (86)					
N	G	12 (66/100/125)					
N	H	25 (36/65/256/303)	10 (90)		2 (269)	4 (134/173)	
N	I				1 (270)	1 (173)	
N	K	2 (100/209)			1 (169)		
N	L	1 (100)					
N	Q	141 (100/209)					
N	S	27 (100/103/117/181)				3 (98/272)	
N	T	1 (66)				1 (168)	
N	W	1 (209)					
N	Y	1 (115)	1 (200)				
P	A	3 (29/178/194)		3 (123)		1 (143)	
P	G				1 (268)		
P	H	1 (178)					
P	K	105 (99)	23 (91)				
P	L	3 (178/188)	1 (269)		3 (156/268/284)		
P	N		4 (91)				
P	Q	95 (156/178/185/285)					
P	S	18 (178/283)	1 (170)		7 (18/27/190/236/243/284)	1 (143)	
P	T	29 (29/178/283)		1 (123)			
Q	A	77 (11/165/220)					
Q	D	1 (199)					
Q	E	35 (165/284)			1 (165)		
Q	H	1 (68)	23 (34/156)		2 (223/293)	1 (88)	
Q	I	1 (68)					
Q	K	2 (11/223)		2 (94/183)		38 (4/37)	
Q	L	3 (11/165/199)	4 (156/270)	1 (129)			
Q	P			1 (116)	2 (175/214)	1 (203)	
Q	R	198 (11/68/239)	27 (35/156)	1 (158)	1 (37)	3 (97/204)	
Q	S	128 (33/51/165)					
Q	T	10 (165)					
Q	V						
R	A					4 (176/271)	
R	C				1 (72)	9 (162/241)	
R	G	2 (10/175)			2 (232)	2 (118/241)	
R	H	25 (10/271/291)	9 (43/68/194)		5 (54/72/219)	22 (162/238/241)	
R	K	131 (10/50/105/208)					
R	L	37 (10/72/195)			7 (76/175/222/232/290)	4 (241/271)	
R	M		1 (97)				
R	N		1 (43)				
R	P	73 (76/105)			1 (175)	1 (41)	
R	Q	74 (10/50/208/221/290)			4 (257/307)	6 (202/271)	
R	S	4 (10/105/175)	10 (43/225)		17 (54/109/219)	37 (63/162/241)	
R	T					1 (41)	
R	V	2 (72)					
S	A	110 (52/67/111/296)	45 (25/89/103)	1 (280)			
S	C	1 (245)	1 (4)			1 (221)	
S	D					4 (2)	
S	E			2 (280)			
S	F				1 (12)		
S	G	101 (111/134/141/158/237/245)	39 (143/231)	2 (18)	2 (36/144)	7 (128/264)	
S	H	2 (289)					
S	I	3 (134/254)	1 (192)		2 (286)		
S	K	36 (158/296)					
S	L				1 (220)		
S	N	6 (26/237/254/289)	29 (4/29/157/275)		3 (81/213)	1 (122)	
S	P					1 (104)	
S	Q	1 (97)					
S	R	83 (5/237/289)					
S	T	275 (67/97/129/141)	37 (4/44/174)	10 (12/37/121/135)	1 (117)	3 (128/233)	2 (131)
S	V	2 (219)	2 (89)				
T	A	192 (13/17/19/21/34/144/206/211/226/281/302)		11 (13/165/293)	9 (16/69/82/252)		9 (25/176/219)
T	C	81 (19)					
T	D	1 (233)					
T	G	3 (17/19)			1 (178)		
T	H	2 (226/233)					
T	I	6 (176/182/226/278)	48 (9/61)		4 (152/212)	1 (267)	
T	K	3 (34/170/211)		3 (202)		1 (186)	
T	M	134 (13/17/226)	8 (168)		1 (283)	19 (261)	10 (104)
T	N	1 (176)			2 (16/212)	1 (186)	
T	P	90 (19/21)			1 (16)	1 (112)	
T	S	130 (21/127/170/179/182/192/193/206)	34 (9/184)	3 (95/221)	4 (129/160/283)		
T	V	108 (144/206)	1 (61)	3 (24)			
V	A	98 (27/85/106/248)	20 (161)	1 (290)	2 (86/91)		2 (19)
V	E					1 (78)	7 (19)
V	F	1 (293)		3 (9)	2 (153)	1 (258)	
V	G	3 (37/247)					
V	I	175 (2/20/37/114/159/293/301)	2 (261)	2 (235)	1 (130)	6 (82)	
V	L	22 (20/91)	11 (17/94)	3 (17)	1 (86)		
V	M	35 (2/37)			5 (86/91/95/277)		
V	P	1 (27)					
V	Q	2 (27/106)					
V	R						
V	S	2 (225/247)				1 (42)	
V	T	68 (247)					
W	C					1 (163)	
W	G					2 (163)	
W	L					1 (163)	
W	P				1 (176		
W	R	1 (246)			1 (71)	5 (163)	
Y	C	10 (31)					
Y	F	1 (31)			8 (5)	1 (103)	
Y	G						
Y	H	22 (31)	1 (144)	4 (115)	2 (5/108)		
Y	N					1 (103)	
Y	S	1 (31)				1 (103)	
Y	W	1 (71)					

Total numbers of mutations: CTX-M: 6344, OXY: 904, PER: 162, SHV: 418, TEM: 534, VEB: 61. Numbers of amino acid insertions: CTX-M: 6, OXY: 10, PER: 0, SHV: 9, TEM: 0, VEB: 0. Numbers of amino acid deletions: CTX-M: 3, OXY: 8, PER: 0, SHV: 2, TEM: 3, VEB: 1. * Total numbers of analyzed sequences: CTX-M (216), OXY (48), PER (15), SHV (199), TEM (199), VEB (27).

**Table 3 antibiotics-10-00812-t003:** Specifications of primers used in the study.

Target (Subtype)	Primer Name	Sequence (5′ to 3′ Direction) ^a^	Natural (N) or Acquired (A)/Phenotype	Amplicon Size (bp)	Tm (°C)	Reference
ACI (1)	ACI-F/R	CCGTTGACATGGAGAATGG, GCGTGTCGGTTATGGAATT	N/ESBL	507	54	This study
ADC (178)	ADC-F1/R1	MAACCTAAAAACYCAATCGGTG, YGGATAAGMAAACTCTTCCCA	N/AmpC, ESBL or CARBA	417–420	58	[34]
ADC (29)	ADC-F2/R2	RGGTTTCTAYCAAGTCGGYA, GCGTTCTTCATTBGGAATACGT		268	59	
ADC (5)	ADC-F3/R3	TGGTCTACAATCCGTTCAAGA, GCCGGGGTTAACTCGAAT		517	54	
ADC (7)	ADC-F4/R4	TATRATGTGCCGGGTATGG, RTCTGTTTGTACTTCAYCTGG		318	54	This study
BEL (4)	BEL-F/R	CGTTCCTTGAAGAGTACGC, ACCCGTTACCCATGAATCA	A/ESBL	401	53	This study
BES (1)	BES-F/R	ATAAGCGGGTGCATTATGC, CTTTAAGCCAGCTCACCAG	A/ESBL	363	53	This study
BPS (11)	BPS-F/R	GCTYCAGTACAGCGACAAC, GTCKTGTTGCCGAGCATCCA	N/cephalosporinase or ESBL	270	57	This study
BPU (1)	BPU-F/R	AAGAAAAGTCCCCATGGGT, CGAACTTGTTCGATGGGAG	N/ESBL	364	55	This study
CARB (8)	CARB-F2/R2	GGGAAAACGTTGGGAACAT, TAATAGCACGCGACCCATA	N or A/BSBL or ESBL	578	54	
CDD (2)	CDD-F/R	AACAAGTGCAAACAATGGC, TTCCTTTACCTTTGGCCCT	N/ESBL	266	52	This study
CdiA (2)	CdiA-F/R	CGTGCTCGCTTTCTTTACT, CACCTGCTCCGGTTTTATC	N/penicillinase or ESBL	692	53	This study
CepA (6)	CepA-F/R	AGTGACAATAATGCCTGCG, TGCTTCGGAATCTTTCACG	N/ESBL	438	52	This study
CfxA (13)	CfxA-F/R	GAAATTGGTGTGGCGGTTA, CAGCACCAAGAGGAGATGT	N or A/BSBL or ESBL	442	53	This study
CGA (1)	CGA-F/R	AGCTACAGTCGGTGTTTCT, TTCATTTTCTGCGCCTGTT	N/ESBL	640	53	This study
CIA (4)	CIA-F/R	GATGGTTTCTGCCTTTGCT, CTTCCGGAAATTTTTCGCG	N/ESBL	299	53	This study
CME (3)	CME-F/R	CCAAAGTGACAACAACGGA, TCCTGAATCGTTCTCAGCA	N/ESBL	376	53	This study
CSP (1)	CSP-F/R	TCTGCTGAGGTTGATTGGA, TCCCACATCATTGGTAGCA	N/ESBL	346	53	This study
CTX-M (208), KLUB (2), KLUG (5), KLUY (5)	CTX-M-F1/R1	ATGTGCAGYACCAGTAARGT, TGGGTRAARTARGTSACCAGA	N or A/ESBL or CARBA	593	55	[37]
CTX-M (7)	CTX-M-F2/R2	ATGTGCAGYACCAGYAAAG, GGCCARATCACCGCRATAT		551	56	[34]
CumA (3)	CumA-F/R	ATCTCCAATGCTATGGGCT, TCACGAGGATCACCATGAA	N/BSBL or ESBL	483	53	This study
DES (1)	DES-F/R	GTTCCAGTTATTCCAGGCG, TGCCAGCACTTTAAAGGTG	N/ESBL	269	53	This study
ERP (1)	ERP-F/R	GTATCGGGCTGTCTCTGAT, GCTGTGCTGTCTGTAATCC	N/ESBL	477	54	This study
FAR (1)	FAR-F/R	CTGAAGAAATCTGGTCGCC, AGCAGTTTCAGGATCTGGT	N/ESBL	473	53	This study
FONA (8)	FONA-F/R	CCGATCTGGTCAACTACAAC, CCCTTCATCCATTCAACCAG	N/ESBL	340	55	This study
GES (45)	GES-F1/R1	ACGTTCAAGTTTCCGCTAG, GGCAACTAATTCGTCACGT	N or A/ESBL or CARBA	624	53	[25]
GES (1)	GES-F2/R2	ATGATCGTGGAGTGGAGCCC, AAGAAGCCGATGTCGTTGCG		448	58	This study
HMB (1), KHM (1)	HMB, KHM-F/R	AAATCGAAGCYTTTTATCCGGG, TTTCCAGCAGCGATGCRTCG	N or A/ESBL or CARBA	237	60	This study
KLUA (12)	KLUA-F/R	CGCTCAATGTTAACGGTGA, TTCATGGCAGTATTGTCGC	N/ESBL	395	52	This study
KLUC (6)	KLUC-F/R	CGATTGCGGAAAAACATGT, CGCCGAGGCTAAWACATC	N or A/ESBL	521	53	This study
KPC (52)	KPC-F/R	CGCTAAACTCGAACAGGAC, CGGTCGTGTTTCCCTTTAG	N or A/ESBL or CARBA	548	54	[34]
LUT (6)	LUT-F/R	TGCTCATGAAAAAGCTGGG, ACCTGTCTTATCGCCTACC	N/BSBL or ESBL	299	54	This study
L2 (18)	L2-F1/R1	TTCCCGATGTGCAGCAC, TTGCTGCCGGTCTTGTC	N/ESBL	518	53	[34]
OHIO (1)	OHIO-F/R	CTTTCCCATGATGAGCACC, CCCGCAGATAAATCACCAC	A/ESBL	599	54	This study
OXA-1-like (11)	OXA(1)-F/R	TCTGTTGTTTGGGTTTCGC, TCTATGGTGTTTTCTATGGCTG	N or A/NSBL, BSBL or ESBL	245	53	[27]
OXA-2-like (22)	OXA(2)-F/R	GATAGTTGTGGCAGACGAAC, TCCATYCTGTTTGGCGTATC	N or A/ESBL	603	55	This study
OXA-10-like (38)	OXA(3)-F/R	ACAAAGAGTTCTCTGCCGAA, TCCACTTGATTAACTGCGGA	N or A/ESBL	418	53	This study
OXA-18	OXA(4)-F/R	ACCATCTGGCTGAAGGATT, CAGAAGTTTTCCGACAGGG	A/ESBL	506	54	This study
OXA-23-like (41)	OXA(5)-F/R	ACTAGGAGAAGCCATGAAGC, ATTTTTCCATCTGGCTGCTC	N or A/ESBL or CARBA	369	55	[34]
OXA-45	OXA(6)-F/R	GCGGTAAACACACTGTCAT, GGGTCAATTGCTGCGAATA	A/ESBL	333	52	This study
OXA-48-like (38)	OXA(7)-F/R	ACCARGCATTTTTACCCGCA, GGCATATCCATATTCATCGC	N or A/ESBL or CARBA	538	55	This study
OXY (28)	OXY-F1/R1	TAAAGTGATGGCYGCCGC, RTTGGTGGTGCCGTAATC	N/ESBL	517	54	This study
OXY (20)	OXY-F2/R2	CCCTGCCTTTATTGCTCTG, TTTATCTCCCACGACCCAG		665	54	
PER (12)	PER-F1/R1	CTCGACGCTACTGATGGTA, TTCATTGGTTCGGCTTGAC	N or A/ESBL	820	54	This study
PER (3)	PER-F2/R2	CTGTTAATCGTGCTGCAGT, GACAAATACCGCCACCAAT		530	53	
PME (1)	PME-F/R	GATCCACTTCAGCGATGAC, GACATCGTGGGTCTTGTTC	A/ESBL	478	54	This study
RAHN (2)	RAHN-F/R	ATGACGTCAGTTCAGCAAC, CATCCATTCCACCAGTTGC	N/ESBL	555	54	This study
RSA1 (1)	RSA-F1/R1	TCGACGATCCTCACTGTTT, GTTGGTGTTCAAATCGGGT	N or A/ESBL or CARBA	483	53	This study
RSA2 (1)	RSA-F2/R2	TGTGGACCTTTCCGAAGAA, CGCGATCAGATTACGAGTG		475	55	
SGM (7)	SGM-F/R	CATGTGCTCGACCTTCAAG, ATCGGCAGCARCAGRTTGG	N/ESBL	225	58	This study
SHV (199)	SHV-F/R	TGGATGCCGNTGACNAACAGC, NTATCGGCGATAAACCAGNCC	N or A/BSBL, ESBL or CARBA	451	59	[34]
SMO (1)	SMO-F/R	CTCACAGACCGTATACCGT, GAATGTCTCATCGCCGATC	N/ESBL	316	54	This study
TEM (199)	TEM-F/R	CACCAGTCACAGAAAAGCA, AGGGCTTACCATCTGGC	N or A/BSBL or ESBL	450	54	[34]
SFO (1)	SFO-F/R	CTCGAGAAAAACTCCGGTG, GTTAGGGTTTGCAGGCTTT	A/ESBL	473	54	This study
TLA (2)	TLA-F/R	GCTAAAGGTACGGATTCGC, CTTAACGCCAAGCTTGCTA	A/ESBL	417	54	This study
TLA2 (1)	TLA2-F/R	ATCGTGCTTGCTGTTTTGA, TCATTTGCCGCATTGTTCT	A/ESBL	623	52	This study
VEB (27)	VEB-F/R	TTTCCGATTGCTTTAGCCG, CCCCAACATCATTAGTGGC	N or A/ESBL	553	54	[34]
YOC (1)	YOC-F/R	CCGGCATCAGAAGAGAAAAA, GGATTCGGGTAGCTTTTGTT	N/ESBL	467	54	This study

^a^ For degenerate primers: B = C or G or T; K = G or T; M = A or C; N = any base; R = A or G; S = G or C; W= A or T; Y = C or T. Abbreviations: AmpC—AmpC, ampicillin chromosomal cephalosporinase; BSBL—broad-spectrum β-lactamase; CARBA—carbapenemase; CTX-M, cefotaxime-hydrolyzing β-lactamase–Munich; ESBL—extended-spectrum β-lactamase; GES, Guiana extended-spectrum β-lactamase; KPC, *Klebsiella pneumoniae* carbapenemase; NDM, New Delhi metallo-β-lactamase; NSBL—narrow-spectrum β-lactamase; OXA, oxacillin carbapenemase/oxacillinase; SHV, sulfhydryl variant of the TEM enzyme; TEM, Temoneira class A extended-spectrum β-lactamase.

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
