# Peer review of "In Silico Analysis of Extended-Spectrum β-Lactamases in Bacteria"

_antibiotics, 2021, doi:10.3390/antibiotics10070812_

Round 1

Reviewer 1 Report

Well writen manuscript in a very relevant topic. The abstract is concise and adequately summarizes the article content. The Introduction represents an adequate synthesis of the thematic. The sections are well organized and include the relevant topics. 

1. The authors should review the Figure 2 considering that is not readable.

Author Response

Reviewer #1:

  1. The authors should review the Figure 2 considering that is not readable.
  • Answer: Thanks to the reviewer for reporting this comment. The resolution of the Figure 2 has been changed.

Reviewer 2 Report

In the submitted manuscript under consideration for publication, the authors compare several ESBL enzymes as well as design primer pairs for their detection. The work was conducted entirely in silico, which is sufficient for the current study. However, it is this reviewer's hope that the authors plan to follow this up with empirical in vitro analysis to validate their primer pairs in the ability to discriminate the ESBL enzymes. The validation is critical for monitoring the presence, prevalence, and spread of ESBL-producing isolates, not only in the clinical setting, but likewise in the agricultural and environmental settings. The presented results are likely of high interest to readers, not only for detection purposes, but also for medicinal chemists potentially designing small molecule inhibitors that target ESBL enzymes.

I have no major concerns with the submitted manuscript in its present form.

Minor concerns:

  1. It is highly recommended that the authors provide full designations for all abbreviations upon the first appearance in the text. Alternatively, the authors could provide a table that lists all the abbreviations and their full spelled designation for ease of reference for readers.
  2. It is unclear the utility of Fig. 1. In its current form, the figure simply lists all the genera in which the ESBL enzymes were detected. Of what significance is the color gradient? This needs more detail, otherwise a table would be sufficient for listing the genera.
  3. Fig. 3 is so small it is nearly meaningless. In print, it is impossible to read the text within the figure. Even zooming in 800% in the digital version, the text is still practically illegible. Highly recommend an alternative format.

Author Response

Reviewer #2:

  1. It is highly recommended that the authors provide full designations for all abbreviations upon the first appearance in the text. Alternatively, the authors could provide a table that lists all the abbreviations and their full spelled designation for ease of reference for readers.
  • Answer: Thanks to the reviewer for reporting this comment. Abbreviations have been added to the text.

  1. It is unclear the utility of Fig. 1. In its current form, the figure simply lists all the genera in which the ESBL enzymes were detected. Of what significance is the color gradient? This needs more detail, otherwise a table would be sufficient for listing the genera.
  • Answer: The data were transferred to the table.

  1. 3 is so small it is nearly meaningless. In print, it is impossible to read the text within the figure. Even zooming in 800% in the digital version, the text is still practically illegible. Highly recommend an alternative format.
  • Answer: Thanks to the reviewer for reporting this comment. Figure 3 was modified.

Reviewer 3 Report

The Authors submitted a paper regarding an in silico analysis of extended-spectrum ß-lactamases in different bacterial genera. Specifically, they investigated whether certain primers could be useful in identifying specific ESBL genes. The work is well written and provides a useful scientific contribution to the topic. The manuscript, after minor revision, could be suitable for publication in the Journal. Please, find attached my comments.

Author Response

Reviewer #3:

  1. I suggest to the Authors to add in the manuscript a list of the abbreviations.
  • Answer: Thanks to the reviewer for reporting this comment. List of the abbreviations has been added to the manuscript.

  1. This Figure is unclear. The Authors should create a single Table reporting all the information (two columns: bacterial genera and ESBL enzymes).
  • Answer: Thanks to the reviewer for reporting this comment. Table was created.

  1. Figure resolution must be improved.
  • Answer: Thanks to the reviewer for reporting this comment. The resolution of the Figure 2 has been changed.

  1. Figure resolution must be improved?
  • Answer: Thanks to the reviewer for reporting this comment. Figure 3 was modified.

  1. Figure resolution must be improved?
  • Answer: Thanks to the reviewer for reporting this comment. The resolution of the Figure 4 has been changed.

  1. Figure resolution must be improved?

Answer: Thanks to the reviewer for reporting this comment. The resolution of the Figure 5 has been changed.

Reviewer 4 Report

Patrik et al. enumerated the sequence of bla genes and designing their primers. Authors need more efforts in this study. Especially, authors must verify the designing primers in Table2 (Authors carry out the PCR and check specificity and sensitivity).

・Fig1 is futility and should change the styles as Table or delete.

・Table 1 and its related manuscript are only showed the mutation of amino acid, and not showed the mutation of position. I think combination of mutation of amino acid and position is important information, however it is uncertain in this manuscript.

Author Response

New things which are added, as per the reviewer’s suggestions, are highlighted in yellow colour in the manuscript.

Reviewer #4:

  1. Patrik et al. enumerated the sequence of bla genes and designing their primers. Authors need more efforts in this study. Especially, authors must verify the designing primers in Table2 (Authors carry out the PCR and check specificity and sensitivity).
  • Answer: Thanks to the reviewer for reporting this comment. Some of our primers have already been verified in other publications (Mlynarcik et al., 2020; Kolar et al., 2021; Mlynarcik et al., 2021), but the remaining proposed primers cannot be tested as we do not have relevant positive controls. Therefore we have performed an in silico hybridization of designed primers with all known ESBL genes described in the BLDB database which can be useful for laboratories that do not have next-generation sequencing platforms and/or do not have sufficient financial resources. The prevalence of ESBL-producing  bacteria is of great concern. Therefore, continuous investigation seems essential to monitor ESBL-producing bacteria in patients with nosocomial and community-acquired infections.

  1. Fig1 is futility and should change the styles as Table or delete.
  • Answer: Thanks to the reviewer for reporting this comment. The data were transferred to the table. We think that the Table provide a more detailed view of the occurrence of selected ESBL genes in bacterial genera. These facts could be used to screen the presence of ESBL in the selected bacterial group.

  1. Table 1 and its related manuscript are only showed the mutation of amino acid, and not showed the mutation of position. I think combination of mutation of amino acid and position is important information, however it is uncertain in this manuscript.
  • Answer: Thanks to the reviewer for reporting this comment. Table 1 was modified.

Round 2

Reviewer 4 Report

.